# Exploiting *V*-Gene Bias for Rapid, High-Throughput Monoclonal Antibody Isolation from Horses

**DOI:** 10.3390/v14102172

**Published:** 2022-09-30

**Authors:** Constantinos Kurt Wibmer, Poppy Mashilo

**Affiliations:** National Institute for Communicable Diseases, Johannesburg 2131, South Africa

**Keywords:** antibody discovery, immunotherapeutics and biologics, immunoglobulin repertoire, antigen-specific B-cell sorting, equine humoral immune system

## Abstract

Horses and humans share a close relationship that includes both species’ viromes. Many emerging infectious diseases can be transmitted between horses and humans and can exhibit mortality rates as high as 90% in both populations. Antibody biologics represents an emerging field of rapidly discoverable and potent antiviral therapeutics. These biologics can be used to provide passive immunity, as well as blueprints for the rational design of novel active vaccine antigens. Here, we exploit the limited diversity of immunoglobulin variable genes used by horses to develop a rapid, high-throughput monoclonal antibody discovery pipeline. The antibodies isolated from two horses in this study were developed with near exclusivity from a few highly related germline genes within a single IgHV and IgλV gene family and could be recovered for cloning with just three primer pairs. This variable gene pairing was compatible with both horse and human immunoglobulin G isotypes, confirming the suitability of an equine antibody discovery pipeline for developing novel therapeutics to meet the One Health approach to infectious diseases.

## 1. Introduction

The horse was likely first domesticated between four and six thousand years ago and has since elegantly bridged the gap between working and companion animal [1]. This co-evolution with humans has significantly shaped the species’ genome, resulting in the selection of genes affecting speed, strength, physiology, development, and behaviour [2]. The close association between our two species has also fundamentally shaped human health. Historically, the horsepox virus (now extinct) is the most likely ancestor of the vaccinia virus vaccine, which was used to eradicate smallpox in humans by 1980 [3]. Equine plasma from hyperimmunised horses has been used for more than a hundred years to make snakebite antivenom which is needed to treat more than 2.5 million venomous bites every year, one fifth of which result in severe deformity, disability, or death [4]. This close relationship has also shaped our two species’ viromes. Equine coronavirus (ECoV) and human coronavirus (hCoV) OC43 (as well as bovine-like coronaviruses) likely share a common ancestor [5]. More recently, equine and human viromes have converged again with the discovery of several emerging pathogens such as Eastern, Western and Venezuelan equine encephalitis, Japanese encephalitis, and West Nile, Nipah and Hendra viruses [6,7,8]. Understanding the equine immune response to these pathogens will help us to design better vaccines and therapies for both horses and humans.

To this end, several studies have focused on characterising the immunoglobulin repertoire of the horse [9,10,11,12,13,14,15]. Antibodies are composed of heavy (IgH) and light (Igλ or Igκ) chains, which can be further divided into constant and variable domains. Antigen recognition is primarily mediated by the complementarity determining regions (CDRs) [16,17]. Diversity within CDRs is achieved through the recombination of germline variable gene segments (IgHV, IgλV, IgκV) with diversity (IgHD) and joining (IgHJ, IgλJ, IgκJ) gene segments. In the horse, there are now 21 known and predicted functional IgHV genes, 9 potentially functional open reading frames (IgHV^ORF^), and 74 IgHV pseudogenes, which can be recombined with 16 predicted functional IgHD genes and 28 potentially functional IgHD^ORF^, as well as 6 predicted functional IgHJ and 3 IgHJ^ORF^ [18]. While there is no agreed standardisation and these genes have been renamed at least three times, a summary can be found on the international ImMunoGeneTics information system^®^ and in the associated literature [11,14,15,19]. More than 90% of circulating horse antibodies use the Igλ light chain [20], which can be formed from recombination of approximately 144 IgλV and 7 IgλJ genes [15]. As of October 2022, these Igλ genes have not been curated in the IMGT^®^ database.

Antibodies can be repurposed as immunoglobulin therapeutics and have been developed as safe and highly effective pre- or post-exposure prophylactics for SARS-CoV-2, Ebola, respiratory syncytial virus, anthrax, and *Clostridioides difficile* [21]. Most of these antibodies have been isolated from infected humans, yielding fully human monoclonal antibodies. The speed of these discovery platforms has increased significantly over recent years, facilitating rapid therapeutic discovery as a component of pandemic preparedness [22,23]. A similarly efficient pipeline for the isolation of antibodies from horses would be of significant benefit, allowing for the rapid discovery of therapeutics to deadly equine pathogens such as African Horse Sickness (which has >90% mortality in naïve populations) [24], or emerging zoonotic pathogens infecting horses and humans [6,7,8].

Here, a rapid 14-day equine immunoglobulin discovery platform was developed. The high-throughput protocol was simplified using just two primer pairs that cover ~90% of IgH*V*- and Igλ*V*-genes used by the horse. A large fraction of the isolated antibodies derived from variable genes previously classed as an ORF or not recognised in older nomenclature. The expressed antibodies were natively paired, compatible with both equine and human IgG isotypes, and are therefore translatable into safe therapeutics for both equine and human use.

## 2. Materials and Methods

### 2.1. Donor Horses

Two adult thoroughbred geldings enrolled in the South African Vaccine Producers (SAVP of the National Health Laboratories, South Africa) antivenom manufacturing program were selected for this study. The animals were chosen for their high-titre plasma antivenom elicited by immunisation, as well as the detection of low-level antibodies to ECoV strain NC99, possibly resulting from a prior exposure. Whole blood was collected from the horses (aged 5 and 8 years old) in CPDA-1 blood bags, and the peripheral blood mononuclear cells (PBMC) were isolated with SepMateTM isolation tubes and Lymphoprep™ density gradient medium using manufacturer recommended protocols (STEMCELL Technologies). Whole PBMC were cryopreserved in foetal bovine serum (FBS) containing 10% dimethyl sulfoxide (DMSO) at 1 × 10^7^. Ethics waiver 20191129-CKW/O was obtained from the Animal Research Ethics Committee, of the University of the Witwatersrand, South Africa.

### 2.2. Antigen Preparation

The ECoV strain NC99 C-terminal domain and subdomain 1 (spike positions 316 to 687) were subcloned into pCMV/R with a synthetic signal peptide (MRPTWAWWLFLVLLLALWAPARG), and a C-terminal HRV3C cleavage site (LEVLFQ/GP) followed by an 8× HisTag and an AviTag (LNDIFEAQKIEWHE). The protein was expressed in Expi293 cells (ThermoFisher SCIENTIFIC, Waltham, MA, USA) and purified sequentially by Ni Sepharose excel beads (Cytiva, Marlborough, MA, USA) and gel filtration. The AviTag was then biotinylated using BirA biotin ligase (Avidity, Aurora, CO, USA) and cleaned of excess biotin by dialysis. Whole lyophilised venom from *Dendroaspis polylepis* was resuspended at 50 mg/mL, biotinylated at primary amines with EZ-Link™ Sulfo-NHS-LC-LC-Biotin (ThermoFisher SCIENTIFIC, Waltham, MA, USA) by manufacturer recommendations and purified from excess biotin. Biotinylated *D. polylepis* venom, polyclonal goat anti-horse (Pierce, Waltham, MA, USA), or ECoV-NC99 CTD+SD1 protein was labelled with AlexaFluor488 (AF488), phycoerythrin (PE), or allophycocyanin (APC) fluorescent streptavidin using saturating levels of biotinylated protein.

### 2.3. Flow Cytometry

PBMC was thawed in phosphate-buffered saline (PBS) containing 2% bovine serum albumin (BSA) and stained with LIVE/DEAD™ Fixable Violet stain (ThermoFisher SCIENTIFIC, Waltham, MA, USA) for 30 min at 4 °C. Cells were washed and then stained with anti-horse antibodies or antigen at concentrations between 0.2–1 μg/mL. After washing again, cells were resuspended in 2 mL PBS (2% BSA) and sorted as IgG+ and antigen single or double positive, at one cell per well, into PCR plates containing Rnasin^®^ Plus Ribonuclease Inhibitor (Promega, Madison, WI, USA) and IGEPAL CA-630. At completion, PCR plates were flash frozen in liquid nitrogen and stored at −80 °C.

### 2.4. Primer Design and Reverse Transcription-Polymerase Chain Reaction (RT-PCR)

All primers (Table 1) were designed based on immunoglobulin genes found in the EquCab3.0 reference genome (GCA_002863925.1, SAMN02953672, PRJNA421018). Primer sets EQ-HV and EQ-λ comprise the final high-throughput primer sets designed in this study. EQ-HC primers were used to check IgG isotype, and the EQ-λV-LEADER-17mix was used to fill in missing lambda sequences. EQ-κ primers were used to search for possible kappa light chains, while EQ-HV screening primers were used in the initial setup to look for non-dominant H*V*-gene representatives in bulk sorted cells.

Thawed single B cell samples were reverse transcribed with SuperScript IV in the presence of fresh RNaseOUT, while immunoglobulin genes were amplified from resulting cDNA using Platinum™ SuperFi II DNA Polymerase and confirmed by size on 48-well E-Gel™ 2% Agarose Gels with SYBR™ Safe DNA Gel Stain, according to the manufacturer recommendations (ThermoFisher SCIENTIFIC, Waltham, MA, USA).

### 2.5. PCR Purification and Cloning

Once confirmed, PCR products were purified from 2% agarose gels using Zymoclean 96-well Gel DNA Recovery Kits (Zymo Research, Irvine, CA, USA) and cloned into pCMV/R using GeneArt™ HiFi Gibson Assembly kits (ThermoFisher SCIENTIFIC, Waltham, MA, USA). Cloned products were transformed into XL-10 gold ultracompetent cells (Agilent) before plasmids were extracted using the Zyppy™ 96-well Plasmid Miniprep kit (Zymo Research, Irvine, CA, USA). Both PCR products and plasmid preparations were confirmed by Sanger sequencing using the BigDye™ Terminator v3.1 Cycle Sequencing Kit (ThermoFisher SCIENTIFIC, Waltham, MA, USA). Heavy chain gene assignment was completed using IMGT^®^/V-QUEST [25,26] while lambda gene assignment was carried out manually.

### 2.6. Immunoglobulin Expression and Screening

Expi293 cells (ThermoFisher SCIENTIFIC, Waltham, MA, USA) were prepared in 24-well plates at 240 rpm, 37 °C, 80% relative humidity, 8% CO_2_ and transfected with 1 μg each of sequence confirmed heavy and light chain plasmids. After 3–5 days, the supernatants were harvested by centrifugation at 5000× *g* for 20 min, and further clarified at 14,000 rpm (in a tabletop rotor) prior to screening. High binding ELISA plates (Corning, Corning, NY, USA) were coated with ECoV-NC99 CTD+SD1 or streptavidin at 4 μg/mL, or whole *D. polylepis* venom at 200 μg/mL (the higher concentration was used to account for the numerous protein toxin components). The plates were blocked with 5% fat-free milk powder in PBS, and incubated with two successive 200 μL aliquots of cell culture supernatant at 37 °C for two hours each. After washing, plates were incubated with anti-human Fc conjugated horse radish peroxidase (Merck, Darmstadt, Germany) in blocking buffer for one hour at 37 °C, and detected with a 4:1 ratio of 1-Step™ Ultra TMB-ELISA Substrate Solution (ThermoFisher SCIENTIFIC, Waltham, MA, USA) for 5 min preceding a 1M acid stop solution. Optical density was read at 450 nm. ELISA positives were expressed in larger 30 mL Expi293 cultures per manufacturer guidelines, harvested by centrifugation and 0.22 μm filtration, and purified using protein A affinity chromatography.

### 2.7. Fc-only Expression and Testing

Horse constant domains IgHC1-3 for IgG1–IgG7 were synthesised (Genscript, Piscataway, NJ, USA) and cloned into pCMV/R. From this, IgHC2,3 was cloned behind an immunoglobulin signal peptide (MGWSCIILFLVATATGVHS) and an HRV3C cleavage site followed by an 8× HisTag and expressed in Expi293 according to manufacturer guidelines. Fc-only dimers were then purified by sequential Ni Sepharose excel affinity chromatography (Cytiva, Marlborough, MA, USA) and gel filtration and coated at 4 μg/mL onto high binding ELISA plates (Corning, Corning, NY, USA). ELISA was performed as above, using biotinylated protein A as the primary detection reagent, and streptavidin-HRP for detection.

## 3. Results

### 3.1. Confirmation of V-Gne Bias within Horse Immunoglobulin G

Two donor horses from the SAVP antivenom program were selected for this study based on high venom binding plasma titres (Figure 1A–left), and detectable binding activity for ECoV-NC99 spike antigens (Figure 1A–right). Memory B cells were bulk sorted as live and IgG+ at 100 cells per well into a PCR plate, and subject to *V*-gene specific RT-PCR. Sixteen primers specific for *HV*-gene leader sequences were paired with four reverse primers to yield 64 reactions, repeated over three replicates. These PCR products served as templates for 48 nested PCR reactions per replicate, amplifying a slightly smaller fragment in each instance (Figure 1B,C). The results for these PCRs are summarised into seven IgH*V*-gene groups (Figure 1C). Only reactions with primers specific for IMGT^®^ subgroup IgHV4 (defined previously as subgroup IgHV2 [14,15]) repeatedly amplified equine H*V*-gene fragments in both first and second rounds of a nested PCR, with increased efficiency for smaller gene products (1500 bp, 1270 bp, 730 bp, and 460 bp). IMGT^®^ defined subgroups IgHV1, IgHV4, IgHV5, and IgHV9 (defined previously as subgroups IgHV1, IgHV2 and IgHV4, IgHV5, and IgHV6, respectively [14,15]) also gave nested PCR products, but in only one or two reactions, using reverse primer Eq-IgHJx(SalI)R after Eq-IgG1CH1/x-R. A multiplexed nested PCR using all EQ-HV screening primers (Table 1) together with Eq-IgG1CH1/x-R and then Eq-IgHJx(SalI)R also yielded a strong IgHV gene product, likely driven by IMGT^®^ subgroup IgHV4. For the light chain products, only Igλ gene primers alone or in multiplexed pools reproducibly amplified strong PCR products, while Igκ primers yielded no product. The entire Igλ gene ORF was recovered. Together with previous studies highlighting IgH*V*-gene and lambda-gene usage bias [9,10,12,13], these data confirm that most equine polyclonal antibodies derive from the IMGT^®^ IgHV4 gene cluster (formerly IgHV2), paired with a lambda gene.

### 3.2. V-Gene Bias and Antigen-Specific Immunoglobulin Properties in Two Horses

To investigate *V*-gene bias within natively paired, antigen-specific monoclonal antibodies from the two-donor horse PBMC, memory B cells were single cell sorted as live, IgG+, and antigen+ (venom and ECoV-NC99, as above) into PCR plates, and subject to *V*-gene specific RT-PCR. Natively paired full-length lambda gene sequences and IgH*V*-gene sequences were recovered from >95% of single sorted B cells. Lambda genes were recovered from a single round of amplification, while IgH*V*-genes were recovered by nested PCR.

From 158 single sorted B cells, two thirds of IgH*V*-genes were derived from the IMGT^®^ defined IgHV4-21^ORF^ (*n* = 58, 37%) and IgHV4-22 genes (*n* = 41, 26%) (Figure 2A). Both genes were not assigned in previous repertoire analyses, and their dominant usage by antigen specific B cells is reported here for the first time. This prevalence was followed by IgHV4S1 (*n* = 31, 20%), IgHV4-29 (*n* = 16, 10%), and IgHV4-82 (*n* = 12, 8%) (referred to previously as IgHV2S2, IgHV2S3, IgHV2S4, respectively [15]). Importantly, despite the second largest proportion of equine *V*-genes having high identity to IgHV4-22, none of the isolated sequences included an extremely unusual cysteine at position 38 (Kabat)/position 43 (IMGT^®^), suggesting an important genomic allelic variant of this gene. IgHV4-21^ORF^ is defined as an ORF in the IMGT^®^ database as: “the gene is partial in 5′ owing to gaps”, and these data support reclassification of IgHV4-21^ORF^ as a functional gene. Sorted antibody heavy chains displayed a range of somatic hypermutation, ranging from 1.1% to 20.7% affinity matured (average 6.7%).

Similarly (Figure 2B), the majority of IgλV genes were derived from IgλV8-128 (alternatively IgλV8S2/Vλ17) and IgλV8-122 (alternatively IgλV8S1/Vλ15), with smaller percentages coming from IgλV8-133 (IgλV8S3/Vλ18), IgλV8-24 (IgλV8S7/Vλ25), IgλV8-12 (IgλV8S9/Vλ27), IgλV4-92 (IgλV4S5^ORF^/VλORF1), IgλV8-28 (IgλV8S5/Vλ22), and IgλV8-20 (IgλV8S8/Vλ26) see [13,15]. IgHJ6*01 was preferentially used by VDJ-recombined CDR-H3, and eleven sequences showed high identity to IgHJ6*03/4^ORF^ (Figure 2C, top). Potentially all of the seven described IgG isotypes were detected (IgG4 or IgG7 could not be segregated in these data) (Figure 2C, bottom). IgG5 was only present in low frequencies, while IgG2 was observed as a VDJ-recombined precursor mRNA with intact introns. IgλC5 and IgλC7 dominated light chain constant domain sequences (Figure 2D), though several alleles of IgλC1 were also detected in 19% of sequences (Appendix A). In stark contrast, no significant gene bias was observed for IgHD genes, and 37 of the 44 currently described IgHD genes (including IgHD^ORF^) were predicted amongst the isolated monoclonal antibodies (Figure 2E and Appendix A).

Insertions and/or deletions (Indels) were found in 20% of heavy chain sequences, ranging from -2 to 7 amino acids long, and were almost exclusively located in the CDR-H2 (a two amino acid insertion in CDR-H1 or framework region 1 was found in three and one antibodies, respectively). Indels appeared to be less common in predominant IgHV4-21^ORF^ or IgHV4-22 derived genes (~14% of all antibodies) when compared to IgHV4S1, IgHV4-29, or IgHV4-82 (~29% of all antibodies).

Overall CDR-H3 length showed a bimodal distribution, peaking at 11 and 14 amino acids long by Kabat numbering [16], or 13 and 16 amino acids by IMGT^®^ [27] (Figure 2F). CDR-L3 length was between 8 and 13 amino acids, peaking at 11 amino acids long (Figure 2G). In both instances, CDR3 amino acid composition was skewed towards glycine (G), tyrosine (Y), and serine (S) (Figure 2H, 2I, Appendix A). This included predominant use of glycine at CDR-H3 position 94 (Kabat numbering) or 106 (IMGT^®^ numbering) which was encoded by preferentially used IGHV4-21^ORF^ and IGHV4S1 genes (Figure 2H, third position). Similar to human antibodies, aspartic acid (D) predominated at CDR-L3 position 108 (IMGT^®^)/92 (Kabat). However, unlike human antibodies glutamic acid (Q) was not common at positions 105 and 106 (IMGT^®^)/89 and 90 (Kabat), where glycine (G) and serine (S) predominated, respectively (Figure 2I, second and third positions, respectively).

### 3.3. A High Throughput Cloning Strategy for Expression as Human and Equine IgG Isotypes

Towards a One Health therapeutic discovery platform, isolated antibodies would need to be rapidly cloned into both human and equine expression cassettes to evaluate neutralisation and various antibody effector functions. Focusing on IgHV4 gene bias, primers were designed that included a 5′ overhang complimentary with the mammalian expression vector pCMV/R (Figure 3A, cyan). Interchangeable IgH isotype cassettes were inserted at the multiple cloning site, including IgHC1, IgHC2, and IgHC3 domains of human or equine antibodies (Figure 3A, yellow). A similar cloning strategy was employed for lambda chain PCR products, where primers introduced 5′ and 3′ complimentary overhangs for cloning into pCMV/R by Gibson assembly (Figure 3B). The V-domains of IgH and Igλ are flexibly joined to Ig constant domains, allowing for easy transition between human and equine isotype without drastically affecting binding or neutralisation (Figure 3C). This way, isolated antibodies can be sub-cloned into various IgG isotypes with desired antibody effector function properties specific to the target antigen.

### 3.4. Donor Horses Displayed a High Proportion of Streptavidin Seropositivity

To assess whether *V*-gene bias was influenced by specific antigens, natively paired antibodies were expressed using high-throughput 24-well plates, and then evaluated for binding to the viral antigen ECoV-NC99 CTD+SD1 (representing natural viral pathogen exposure), or *D. polylepis* whole venom toxins (induced through immunisation). A total of 72 of 158 randomly selected heavy-light chain antibody pairs were transfected into 24-well plates with Expi293 cells and cultured for 5 days before assessing supernatant binding by ELISA. Initially, this yielded a relatively low antigen-positive ratio relative to the total sorted B cells (39%). To search for additional unintended sorting antigens, transfected cell culture supernatants were screened against the bacterial antigen streptavidin (used as the antigen fluorescent labelling agent). Remarkably, streptavidin binding antibodies accounted for 42% of the sorted B cells (Figure 4A, blue). Between all three antigens, 80% of the sorted antibodies were antigen specific, with 12 binding to ECoV-NC99, 16 binding to snake toxin, and 30 binding to streptavidin (Figure 4A). Altogether, *V*-gene usage bias was similar between three antigenically distinct groups: viral (ECoV-NC99), toxin (*D.polylepis*), and bacterial (streptavidin).

### 3.5. Equine V-Genes Are Equally Functional as Horse Antibodies and Human IgG-chimeras

The top binder for each antigen group (Antibody 17b for ECoV-NC99, p3A4 for *D. polylepis*, and D3 for streptavidin) was then selected for equine isotype and human chimerisation studies. To first assess ease of purification by protein A, recombinant Fc-dimers of all seven equine isotypes were expressed with a HisTag in isolation and tested for binding to in-house protein A (Figure 4B). Only IgG6 was not bound by recombinant protein A, with detectable binding to all other isotypes. Based these binding data, the relative prevalence of each isotype (Figure 2C), and ability of each isotype to mediate effector functions [28], the three isolated antibodies were subcloned into expression vectors encoding constant domains for human IgG1, equine IgG1, or equine IgG7. These nine antibodies were then tested for binding to the three antigen groups. All isotypes, whether human or equine, retained equivalent levels of antigen recognition measured by ELISA (Figure 4C), confirming the suitability of equine variable domains for human antibody scaffolding (Figure 3C).

## 4. Discussion

Monoclonal antibodies represent a first line in rapidly discoverable antiviral therapeutics for infectious diseases. Herein, a rapid, high throughput protocol for the discovery of monoclonal antibodies from horses has been described. In summary, PBMC were processed from immunised/infected horses and stained with fluorescent antigen, before single B cells were sorted, lysed, and subject to RT-PCR for IgHV and Igλ genes. PCR products were purified by commercial 96-well gel extraction kits, cloned by Gibson assembly, and prepared for transfection using commercial 96-well miniprep kits. The resulting products were used to transfect Expi293 cells in high-throughput 24-deepwell plates, and cell culture supernatants were harvested after five days for screening by ELISA. When compared with other equine mAb discovery platforms [12,29], this protocol is unique in the exploitation of *V*-gene bias to accelerate lead candidate identification (14 days) and allows for the isolation of full-length natively paired antibodies, with the ability to easily switch between human or equine IgG isotypes.

The high prevalence of streptavidin-specific antibodies was unexpected and complicated our antigen-specific sorting strategy. These high titre antibodies may represent environmental exposure to avidin-like protein from various unknown bacterial sources. Based on these data, alternate strategies for the fluorescent labelling of sorting antigens are recommended and will be developed in future studies. In total, the process takes less than 14 days from whole blood to identifying antigen-specific lead candidates for further study (Figure 5).

These methods were simplified by the *V*-gene usage bias of expressed horse antibodies. *V*-gene bias for both heavy and light chain genes of the horse have been shown previously. Initially, Almagro et al. described antibodies from single scorpion antivenom producing horse that all belonged to a single family dubbed IgHV1 [9]. This discovery predated the first comprehensive analysis of the germline immunoglobulin repertoire of the horse, which expanded this family into three main genes: IgHV4, IgHV5, and IgHV14 [11]. These genes were later reclassified as IgHV2S2, IgHV2S3, and IgHV2S4 and shown to dominate expressed antibodies through all stages of life [14], as well as antigen-specific antibodies raised against keyhole limpet hemocyanin or equine influenza neuraminidase [12]. More recently, these genes were reclassified yet again as members of the larger IgHV4 gene family [18]. Based on the most recent IMGT^®^ classification, we confirm here that almost all expressed antibodies use just a handful of IgHV4-related genes (independent of antigen) and show that the majority of these derive from IgHV4-21^ORF^. While our study was limited in sample size (*n* = 2), reanalysis of 920 NCBI GenBank sequences (summarised in [29]) revealed the same gene bias predominated by IgHV4-21^ORF^ (Appendix A). Importantly, none of our antibodies which were assigned to IgHV4-22 had the rare framework region 2 (FWR2) cysteine residue described in the IMGT^®^ data. This could represent a database sequencing error, or an important allelic variant that may impact IgHV4-22 gene usage between different individuals. Furthermore, we discovered several highly mutated heavy chains (17–20% divergent from the closest germline relative) that may represent yet undiscovered germline genes, or highly affinity mature antigen specific sequences.

An overview of the mAb isolation protocol is shown, and average timeframes are given in brackets. This does not account for time to acquire antigens. The entire process takes less than two weeks. Created with BioRender.com.

Similar to heavy chain *V*-gene bias, we confirmed dominant usage of several related IgλV8 genes, as has been previously shown [13]. These lambda genes have not yet been fully catalogued in IMGT^®^ and are likely to renamed. The prevalence and high similarity of IgHV4:IgλV8 paired antibodies in horse PBMC facilitated the development of a high-throughput antibody discovery platform using just three primers pairs, which resulted in >90% immunoglobulin gene recovery. The remaining antibody sequences could be recovered by a more gene-specific targeted approach where necessary, such as the EQ-λV-LEADER-17mix lambda primer pool.

The composition of CDR-H3 in the horse antibody repertoire has recently been interrogated by next-generation sequencing and was shown to have a low conservation of charged amino acids at positions 106 (IMGT)/94 (Kabat) and 116 (IMGT)/101 (Kabat) [10]. Here, we show that the high proportion of glycine at position 106 (IMGT)/94 (Kabat) is directly encoded by the dominantly used IgHV4-21^ORF^ and IgHV4S1, explaining this phenomenon. We also observed an unusually high frequency of glycine, serine, and alanine at CDR-L3 positions 89 and 90, normally occupied by glutamine. The combination of short, flexible amino acids at the base of both CDR-H3 and CDR-L3 is likely to result in greater flexibility resulting in increased conformational sampling of CDR3. In addition, we observed diverse inclusion of IgHD genes in horse CDR-H3, which were ascribed to at least three quarters of the known IgHD gene repertoire. This agrees with next-generation sequence data for CDR-H3 [10] and suggests that despite limited IgHV and IgHλ gene usage, horses are able to recognise numerous antigens through diverse CDR3 loop conformations and compositions.

The isolated IgV genes could be successfully cloned into both human and horse immunoglobulin isotypes, without loss of binding. This experiment was critical to confirming the usefulness of a rapid horse antibody discovery pipeline for the identification of novel biologics to treat emerging threats in both human and equine populations. Further study will aim to apply this pipeline to additional equine and human health priorities, assess the neutralisation and effector functions of isolated antibodies, and understand structural differences between equine and human antibodies to rationally develop fully human monoclonal antibodies using isolated equine templates.

## Figures and Tables

**Figure 1 viruses-14-02172-f001:**
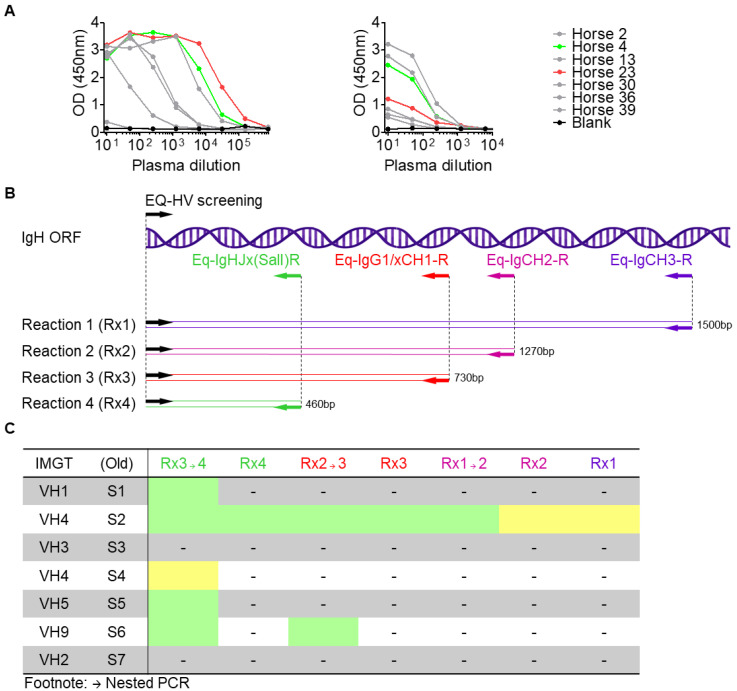
Optimisation of IgHV gene recoveryM. (**A**) ELISA data for horse plasma screened against *D. polylepis* whole venom (left) or ECoV-NC99 CTD+SD1 (right), Absorbance (OD450 nm) is shown on the *y*-axis, and plasma dilution on the *x*-axis. The two horses selected for this study are shown in green and red. (**B**) Location of gene specific primers relative to the heavy chain open reading frame (IgH ORF) is shown. Several IgH*V*-specific primers were designed in the leader sequence (EQ-HV screening) and paired with complimentary reverse primers for amplifying almost the entire heavy chain ORF (Rx1: Eq-IgCH3-R, purple) or smaller fragments by regular or nested PCR that included partial Fc sequence (Rx2: Eq-IgCH2-R, magenta), nearly the entire Fab sequence (Rx3: Eq-IgG1/xCH1-R, red), or the variable domain only (Rx4: Eq-IgHJx(SalI)R, green). (**C**) The results of the IgHV-gene specific PCRs have been summarised by IMGT family (the most recent previous subgroup designation is shown in column 2). Single round and nested PCR reactions are coloured as in A, where nested PCRs are labelled Rx1 → 2, Rx2 → 3, and Rx3 → 4. A single weak PCR positive in any reaction is shown in yellow. Strong and/or recurrent PCR positives are indicated in green.

**Figure 2 viruses-14-02172-f002:**
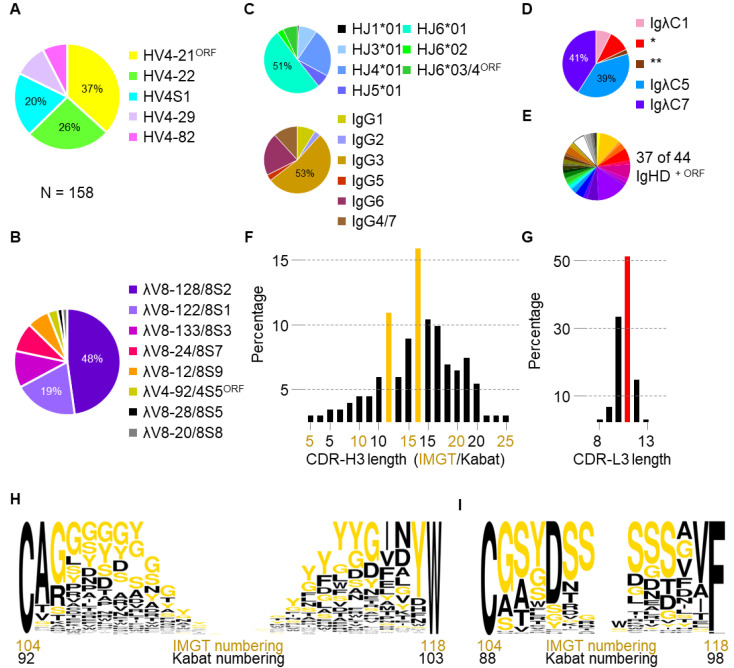
*V*-gene analysis of single sorted B cells. (**A**) Pie chart summary of the IgHV genes used by 158 single sorted B cells. Slices are listed in order of proportion relative to total. (**B**) Pie chart summary of the IgλV genes paired with the same antibodies as A, listed in order of proportion relative to total. (**C**) Summaries of IgHJ genes (top) and IgHC genes used by the same antibodies in A, listed in order of gene number. (**D**) Summary of IgλC genes used by the same antibodies as A. Allelic variants of IgλC1 are indicated with the asterisks, and are shown in Appendix A. (**E**) Summary of IgHD gene usage predicted from the isolated antibody CDR-H3 mature sequences, listed in Appendix A. (**F**) Bar graph showing CDR-H3 length distribution for the isolated sequences. The relative frequency in percentage is shown on the *y*-axis, while CDR-H3 lengths are reported on the *x*-axis. Both IMGT^®^ (gold) and Kabat (black) length reporting is shown. The bimodal peaks in length distribution are indicated with the gold bars. (**G**) Bar graph showing CDR-L3 length distribution, labelled as in F. The peak distribution at more than half of total sequences is indicated with the red bar. (**H**) A CDR-H3 logogram showing the amino acid composition at each position. White spaces represent decreasing representation with increasing CDR-H3 length. Predominant amino acids G, S, and Y are coloured gold. (**I**) Logogram of CDR-L3 amino composition, coloured and labelled as in H.

**Figure 3 viruses-14-02172-f003:**
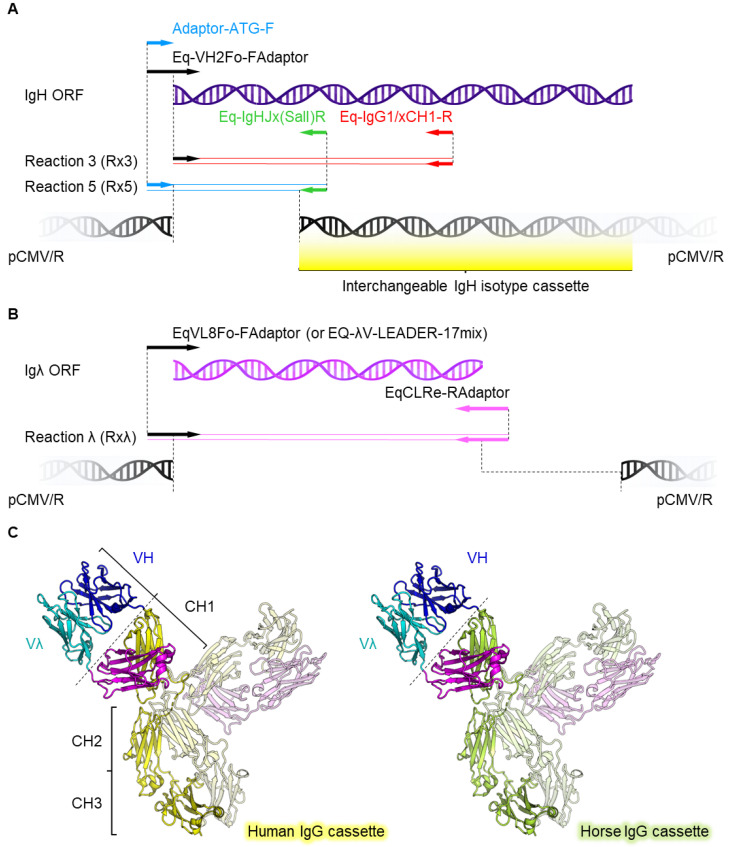
Optimisation of a rapid, isotype exchangeable IgG cloning strategy. (**A**) Schematic of the high-throughput cloning strategy, showing introduction of the adaptor 5′ sequence (cyan), and VH-gene 3′ sequence (green), both of which overlap with cloning sites in the expression vector pCMV/R (black). Gibson assembly using these primer overhangs can be used to clone H*V*-gene fragments into vectors expressing the constant domains for all known human or horse IgG isotypes. (**B**) Whole λ gene cassettes (pink/purple) were similarly cloned. (**C**) Structural schematic of the horse *V*-genes cloned into exchangeable human/equine IgG isotype expression cassettes, each with unique effector function capabilities. The VH/Vλ domain exists at the end of a flexible hinge region (dotted line), and this exchange does not affect antigen recognition or neutralisation.

**Figure 4 viruses-14-02172-f004:**
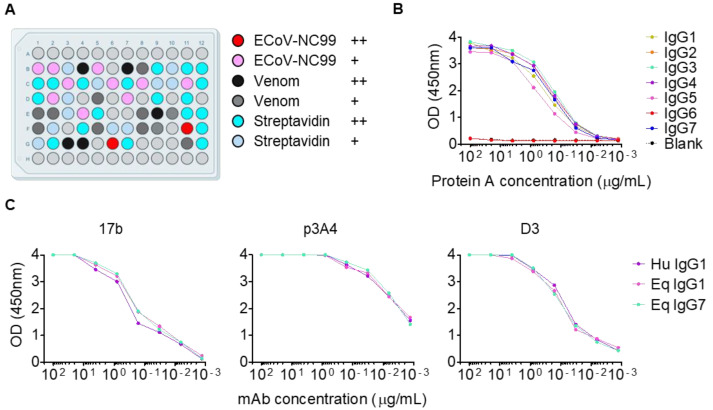
Antigen recognition by human and equine isotypes. (**A**) Single point ELISA plate map of monoclonal antibody cell culture expression supernatants. Absorbance measurements greater than half maximal OD450 nm (++ OD: 4 > 2) for ECoV-NC99, *D. polylepis* venom, or Streptavidin was coloured red, black, and cyan, respectively, while ELISA positives less than half maximal OD450 nm (+ OD: 2 > 0.5) were coloured pink, grey, and light blue, respectively. (**B**) Titrated ELISA of equine Fc domains bound to protein A. Absorbance (OD450 nm) is shown on the *y*-axis, and protein A concentration on the *x*-axis. IgG6 Fc did not bind protein A (red) and overlaps with the blank curve (black). (**C**) ELISA curves for ECoV-NC99 binding antibody 17b, *D. polylepis* binding antibody p3A4, and streptavidin binding antibody D3 are shown. Absorbance (OD450 nm) is shown on the *y*-axis, and antibody concentration on the *x*-axis. Antibodies were cloned into a human IgG1 isotype expression cassette (purple), a horse IgG1 expression cassette (pink), or a horse IgG7 expression cassette (cyan).

**Figure 5 viruses-14-02172-f005:**
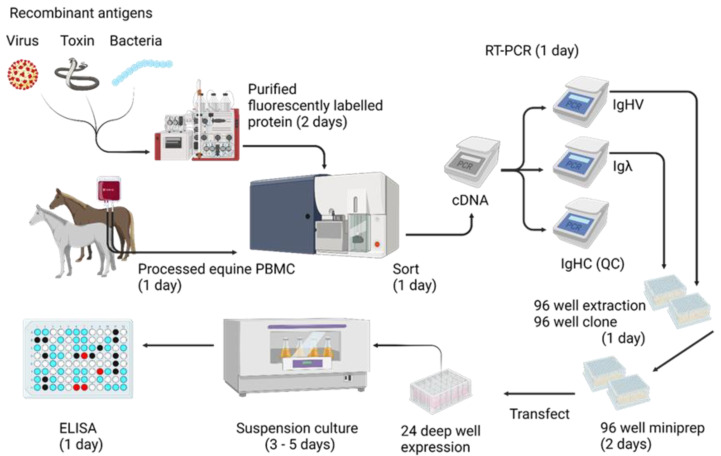
A high-throughput monoclonal antibody isolation pipeline for horses.

**Table 1 viruses-14-02172-t001:** Primer sets used for horse antibody discovery pipeline.

**EQ-HV high-throughput primers (20 μM)**	
Adaptor-ATG-F	TTCGCTCTAGAGCCGCCACCATG
Eq-VH2Fo-Fadaptor	TTCGCTCTAGAGCCGCCACCATGA*A*TCACCTGTGGTTCTTCCTCTTTC
Eq-IgHJx(SalI)R	GTGGTCGACGCTGAGGAGACGGTGACCAGG
Eq-IgG1CH1-R	TGCTGGCCGCGTGGACTACG
Eq-IgGxCH1-R	TGCTGGCCGGGTGGGCTAC
**EQ-HC primers (20 μM)**	
Eq-IgCH2-QCF	GGGCCTTCSGTSTTCATCTTCC
Eq-IgCH2-R	CCAGGCAGGTCACGCTGACC
Eq-IgCH3-R	GAGCTTGCTGTACAGGAAGTAGGACC
**EQ-λ high-throughput primers (5 μM)**	
EqVL8Fo-Fadaptor	TTCGCTCTAGAGCCGCCACCATGGCCTGGTCCCCTCTCCTCC
EqCLRe-Radaptor	GGCACAGCAGATCTGGATCCTCACTAAGGACACTCTGAGGGGGACASTTTC
Adaptor-Stop-R	GGCACAGCAGATCTGGATCCTCACTAAG
**EQ-HV screening primers (20 μM)**	
EqVH1-LEADER-A	ATGGACTGGAGCTGGAGCATCC
EqVH1-LEADER-B	ATGGGCTGGAGCTGGAGAATCC
EqVH2-LEADER-A	ATGAATCACCTGTGGTTCTTCCTCTTTC
EqVH2-LEADER-B	ATGAGTCACCTGTGGTTCTTCCTCTTTC
EqVH3-LEADER-A	ATGGAGTTTGGGCTGAGCTGGATTTTC
EqVH3-LEADER-ORF	ATGGAGTTTGGGCTGATATGGGCTTTTC
EqVH4-LEADER-A	ATGAGACTCTTGTGTCTTCTCCTTTGCC
EqVH4-LEADER-Ax	ATGAGACTCTTGKGTCTTCTCCTTTGC
EqVH4-LEADER-B	ATGAGACTCTTGTGTCTTCTCCTTTTCC
EqVH4-LEADER-C	ATGAGGAGGCTGGGTCTTCTCC
EqVH4-LEADER-D	ATGAGGCTGTTGGGTCTTCTCCTTTG
EqVH4-LEADER(ORF)	ATGAGATTGTTTGGTCTTCTCTTTTGCTTGG
EqVH5-LEADER	ATGGGCTCTGCCACTGAACTTGC
EqVH6-LEADER-A	ATGGCCCCTCTCCTGGTCATCTTCTG
EqVH6-LEADER-B	ATGGCCCCTCTCCTGGTCGTC
EqVH7-LEADER	ATGGACACACTGTATCCCACCCTC
**EQ-** **κ primers (20 μM)**	
EqVk1/2-LEADER-A	ATGAGGSTCCCTGCTCAGCTC
EqVk2-LEADER-B	ATGAGGTTCCCTGCTCAGCTCC
EqVk2-LEADER-C	ATGAGGTTCTCTGCTCAGCTCCTG
EqVk2-LEADER-D	ATGAAATTCCTTGCTCAGCTCCTGGG
EqVk2-LEADER-E	CTAGTCAGCTCCTGGGGCTAC
EqVk2-LEADER-pan	ATACTCTGGATCCCAGGATCCASTG
EqVk4-LEADER-A	ATGCTGACGCAGACGCAGGTC
EqVk4-LEADER-B	ATGATGTCGCTGACACAGTTCCTTATATC
EqVk4-LEADER-C	ATGATGTCATGGACTCAGATCCTTATGTC
EqVk4-LEADER-D	ATGATGTTGCAGACACAGGTCCTTATAAC
EqVk4-LEADER-E	ATGATGTCACAGACACAGGTCCTTATATC
EqVk4-LEADER-F	GATGTCACAGACACAGGTCCTCTTG
EqVk4-LEADER-G	GATGTGGGAGACACAGGTCCTTATG
EqVk4-LEADER-H	ATGATGTCGCTGACAAAGGTCCTTATATC
EqVk4-LEADER-I	GATGTCACTGACAAAGGTGTTTATGTCTTTG
EqVk4-LEADER-pan1	TTGGGTCTCAGGWGCCTGTGG
EqVk4-LEADER-pan2	TCTGGGTCTCAGGWGCCTGTG
EQVK5-LEADER	GGCTCCCAGGCTCAGCTCC
EQVK6-LEADER	ATGGTGTCCCCATCACAGCTCC
EQVK9-LEADER	ATGAGCTTCCAGGCCCAGCTC
EQCK-ROUT	ACAATCTTSCCTWTTGAAGCTCTTGACC
EQCK-RIN	GCTCAGGGTCTTGTGGGAGACC
**EQ-** **λV-LEADER-17mix (6 μM)**	
EqVL1-LEADER-A	GGCCTGGACCCCTCTCCTG
EqVL1-LEADER-B	ATGGCCTGGACCCTTCTCCTG
EqVLorf4-LEADER-A	ATGGCCTGGACTCCTCTCCTC
EqVL9-LEADER-A	ATGGCCTGGACTCCTCTCATCC
EqVL9-LEADER-B	GTCCTGTACTCCTCTCCTCCTC
EqVL4-LEADER-A	ATGGCCTGGACCCCTCTCTTG
EqVL5-LEADER-A	TGGCCTGGACACTTCTCCTTCTC
EqVL6-LEADER-C	ATGGCCTGGACTCTGCTCCTTC
EqVL6-LEADER-A	GGCCTGGGCTCTGCTCCTC
EqVL6-LEADER-B	GCCTGGGCTCTGTTCCTCATC
EqVLorf3-LEADER-A	ATGGCCTGGGCTCCGTTCTTC
EqVL7-LEADER-A	GGCCTGGGTGCCACTCCTG
EqVL8-LEADER-A	GGCCTGGTCCCCTCTCCTC
EqVL8-LEADER-B	GGCCTGGTSCCCTCTCCTC
EqVL10+LEADER-A	ATGGCCTGGACGGTGCTTCTTC
EqVL10+LEADER-B	GGCCTGGATGGTGCTTCTTCTC
EqVL10+LEADER-C	GGCCTGGACAGTGCTTCTTCTC

## Data Availability

All data will be made available upon request from the corresponding author.

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
