# Peer review of "Exploiting V-Gene Bias for Rapid, High-Throughput Monoclonal Antibody Isolation from Horses"

_viruses, 2022, doi:10.3390/v14102172_

Round 1

Reviewer 1 Report

Reviewer summary:  Wibmer and Mashilo show that due to an extreme bias in the variable gene usage in the horse, immunoglobulin can be easily cloned using a limited number of primer pairs.  These variable immunoglobulin genes seem to be compatible with horse IgG1 and IgG7, as well as, human IgG1.  This method may be useful in rapidly identifying therapeutic antibodies that can benefit both species.

General comments:

The databases for the horse immunoglobulin repertoire are in their infancy, based on far too few animals to be considered complete. The authors provide data which will help solidify information about the equine immunoglobulin repertoire. 

Two horses were used to develop the method that were then used to validate the method using limited primer pairs for immunoglobulin discovery.  In the end, this only validates it works with these two horses.  It would improve the soundness of this method to test it using unrelated horses, vaccinated with a different antigen/pathogen.  The authors need to stress that this method may have limitations because of the small sample size.  So much is unknown still.  Different breeds of horse, as well as, various exposure to different pathogens, parasites, and environmental factors may lead to different immunoglobulin variable genes being utilized.  This manuscript does provide important information but the conclusions are too premature to put out into the scientific community without further validation.

Although the authors show that the humanized horse antibodies retain binding by ELISA, this doesn’t necessarily translate to retention of function.  Until tests such as in vitro neutralization assays and/or assays to test Fc effector function (ADCC, ADCP, etc.) are completed, it is difficult to argue that these antibodies would have any value beyond diagnostic capabilities.

Specific line comments:

11-12    “passive vaccine” – better to say they provide passive immunity.  Vaccine or immunization implies the receiver of the antibodies will be stimulated to produce antibodies.

57          “vaccine” is an inappropriate term here, again, because it implies that antibodies are administered to produce more antibodies.   

75          Please characterize these horses more – Breed, antigen exposure, etc.  The authors add more information in lines 308-309 but that is not the correct place for that.

135-136              The levels of antigen coating on the plates seems really high.  For proteins, we typically coat plates between 0.5-2.0 ug/ml.

Figure 1A            The arrows and colored bars would seem to indicate only that portion of the IgG was amplified.  I think you are trying to represent that you amplify from the right colored bar to the black colored bar (EQ-HV Screening) for each

Figure 1B             In the figure description, define Rd1, Rd2, and Mx.  It may also be better to reorient the columns to coincide with Figure 1A.

Figure 2                              The logograms in H and I are too compressed to be able to read the letters for all the amino acids.  It may be better to show each at the full width of the page.

Reviewer 2 Report

This paper describes a rapid and high throughput protocol for the discovery of horse monoclonal antibodies. It’s a very interesting and important topic since antibodies has been used so much in not only research field but also in public health. It’s important to have the new strategies produced and reported. This paper is rich in content with good flow overall.

However, it is better to add some in the introduction/discussion on how your protocol is simplified and better than old ones. Also, in the introduction part, like line 36-40, and 41-44, needs references. 

Reviewer 3 Report

In this study, Wibmer and Mashilo strived to take advantage of the V-gene bias among horses and humans in order to develop a potential source of monoclonal antibodies.

The study deals with a highly up-to-date topic that deserves attention. The study is well designed and executed, the data are novel and worth of further investigation.

I only have 2 comments for possible improvement:

-        More information regarding the horses would be helpful. What breed and gender were they? Was not the study limited by employing only two animals to obtain the samples for further experiments?

-        The authors could add a paragraph on any potential limitations of their experiments as well as future prospects.

Round 2

Reviewer 1 Report

I am satisfied with the changes made by authors.  

Just one note:  The figure 4 description does not come immediately after the figure.